# Visioning the Future of Smart Fashion Factories Based on Media Big Data Analysis

**Sae-Eun Lee** [1], **Naan Ju** [2,*] **and Kyu-Hye Lee** [1,*]

1    Human-Tech Convergence Program, Department of Clothing and Textiles, Hanyang University, Seoul 04763, Korea; bion0814@hanmail.net
2    Research Institute of Industrial Science, Hanyang University, Seoul 04763, Korea
*    Correspondence: naan_ju@hanyang.ac.kr (N.J.); khlee@hanyang.ac.kr (K.-H.L.)

**Abstract:** Recently, many companies have adopted smart factories to increase productivity and efficiency. However, the fashion industry is one of the industries that have been relatively slow at embracing automation and switching to a smart factory. The purpose of the study is to suggest the future direction of the low-maturity smart factory in the fashion industry through newspaper analysis. In this study, semantic network analysis and convergence of iterated correlation (CONCOR) analysis were performed on 15,523 news articles. The analyses revealed that the smart fashion factory was developing to incorporate automated, unmanned, and intelligent operation. The problem of job loss owing to the smart factory was also heavily addressed in the news articles. In the newspaper articles, the view that the smart factory is efficient, fast, and innovative, and concerns regarding the possible damages that will result from hacking and machine malfunction were simultaneously expressed. Therefore, if news about security improvement emerges in the future, negative public opinion will be reduced, positively influencing the government's support for smart factories and policy making.

**Keywords:** smart factory; smart fashion factory; semantic network analysis; CONCOR analysis

## 1. Introduction

In the fourth industrial revolution (4IR), the smartening of manufacturing companies has become important. This is because the global economy has entered an era of zero growth. Utilizing the Internet of Things (IoT) and artificial intelligence (AI) has become a survival strategy in the fashion industry. Fashion companies have transformed the entire supply chain and achieved quick responses and mass customization through IoT and AI. In addition, companies have aimed at increasing productivity and efficiency through smart factories with robots and machines [1]. In addition, the structure of the new industrial ecosystem and the value chain is switching to a parallel form rather than a serial one [2]. Germany, America, and China are pursuing Industry 4.0, Making America, and Made in China 2025, respectively, in response to the new industrial ecosystem [3].

According to Forbes data on smart factory initiatives in the industry, smart factories have been adopted for 67% of industrial manufacturing, 50% of automotive and transportation production, and 40% of consumer goods production [4]. The smartening of the fashion manufacturing sector is so weak that it is subsumed under "others" [5]. The first smart fashion factory was Adidas' Speedfactory, which was established in December 2015. Since then, Amazon's on-demand apparel manufacturing system, SewBot of SoftWear Automation, the robot-guided sewing process of SINTEF Raufoss Manufacturing AS in Norway, JUKI Smart Solution in Japan, and a seven-day customization process by Kutesmart in China have appeared [6–10]. However, there have only been a few other cases. Overall, smart factories in the fashion manufacturing field are developing very slowly, compared to other industries. This can be attributed to labor intensity and low labor costs that characterize the fashion manufacturing sector, among other factors that are peculiar to

the industry [11]. However, in the future, the fashion field, like other industries, must aim for the development of smart factories. To solve problems such as shortage and aging of skilled workers, response to consumer demand, shortened product lifestyles, and efficient use of resources.

In addition, previous research on smart factories mainly consists of construction and implementation of such key-technologies, performance, solutions, and applications [12–15]. It is necessary to study the technology and advantage of smart factories and investigate them from the public and media's point of view to strengthen future competitiveness in an uncertain business environment and determine the direction of business development from a long-term perspective. Furthermore, relying on the intuitive insight of experienced experts is expensive and has limitations that can provide varying results from participants. As large amounts of information increase exponentially in a short time nowadays, it is not appropriate to rely on expert analysis to capture future opportunities and derive industry competitiveness. Therefore, this study used text mining that can extract high-quality information from various sources. This is to analyze the development status of and change in smart factories and provide basic data for the development of smart fashion factories. Jung and Chang [16] and Kwon and Lee [17] identified topics and trends for online news on smart factories through text mining analysis but limited their subjects to Korean news only. Although there are some studies that have a global focus, they are limited to identifying trends or topics [18]. Beyond understanding topics, there is also the need to grasp industry trends and the atmosphere of public opinion. The news influences the perception of certain issues. Media audiences learn a variety of information exposure to news. Furthermore, their perception of the importance of topics in the news is often determined by how much the news media emphasizes them [19]. In other words, news reports in the media on public issues use a specific frame to influence the formation of citizens' opinions and steer public opinion in a certain direction [20]. News can shape public opinion; therefore, it is necessary to first consider the influence of the information obtained from news on the advancement of the smart fashion factory.

The technological progression of the fashion industry was predicted through text mining. Text mining entails the use of quantitative and qualitative analyses, both of which are used to forecast the future. While predictions of technological growth based on the statistical representations of extracted keyword distributions are quantitative, the interpretations of possible key elements of this growth depend on technical discourse [21]. According to Jasanoff [22], it is fundamentally assumed that all scientific knowledge and technological progress depends on the "co-production" between technological advancement by experts and the recognition of the importance of technology by society. The more often the news is reported, the greater the awareness of the need for the smart factory. News also shapes public opinion and policymaking that helps smart fashion factories grow and spread.

Therefore, in Section 2, we briefly review the smart fashion factory technologies currently used, and in Section 4, we suggest a development direction by examining key topics regarding smart factories that have appeared in the media. In addition, by searching the positive/negative aspects of public opinion, we propose guidelines for attributes and directions when establishing a smart factory in the fashion field. This will empower small-medium fashion companies to build smart factories. These companies do not have enough assets to build smart factories and need government support. The aim of this study is to provide insight into the future of fashion smart factories by analyzing the development direction and changes in smart factories through a big data analysis of newspaper articles. The following questions were studied: RQ1. What are the main issues and attributes of smart factories that have appeared in news articles? Through this, important attributes can be referenced when constructing a smart fashion factory.

RQ2. What are the positive and negative words used in relation to smart factories in news articles? Due to the nature of the fashion field, there are many small-medium companies, and the government's help is urgently needed, so the direction of public

opinion should be carefully considered. This can help to form a positive public opinion, which would be the basis for policy establishment regarding smart fashion factories.

## 2. Theoretical Background

### 2.1. Maturity of Smart Fashion Factory

In this section, we look at the maturity level of smart fashion factories. Most of the fashion industry belongs to small and medium-sized enterprises (SMEs), and some studies confirm maturity in some SMEs [23,24]. This suggests future directions for improvement after examining the low maturity level of smart fashion factories compared to other industries. However, there are some case studies on smart fashion factories [25], but few studies focus on the fashion industry's maturity. The fashion industry's characteristics are slightly different from other SMEs. This is because fashion manufacturing is a complicated manufacturing process because of frequent changes in style; moreover, some styles require intricate manual work. However, the reality is that the fashion manufacturing industry is changing from being labor-intensive to being technology-intensive owing to overall smartization. A smart fashion factory is a fashion manufacturing company that aims to produce " customer-tailored products" [26]. Some fashion manufacturing companies are attempting to minimize the difference be-tween skilled and unskilled workers through automation using ICT technology and intelligent robots [27].

Various models have been proposed to diagnose the level of response in the 4IR. Chonsawat and Sopadang [24] classified the maturity of SMEs into 5 stages according to the level of automation, machine integration, and cloud system. According to Kim and Moon [11], the level of automation in the fashion manufacturing sector is low and slow, compared to other industries, and with the exception of some companies, such as Germany's Speedfactory and China's Kutesmart, most of them remain at the basic level, without ICT application. In addition, in the data reported by the Ministry of Trade, Industry and Energy of Korea in 2017, the initiative rate for smart factories in the fashion sector was 0.1% and is presently unlikely to increase significantly; it is likely to maintain a low level. Therefore, it is necessary to examine what is needed to improve this low maturity of smart factories.

### 2.2. Technologies of Smart Fashion Factory

The review of the technology currently being implemented in smart fashion factories helps infer the need for future improvement. According to the SME maturity model analysis [24], it is necessary to examine the current situation (Table 1) systematically before proposing a future direction for smart fashion factories. There are not many previous studies on smart factory technologies in the fashion industry [28]. Zahra et al. [28] focused on driving factors for application through a qualitative analysis rather than a manufacturing step-by-step approach or analysis of current technology. The fashion manufacturing process can be divided into three areas: development, manufacturing, and supply chain. However, it was observed that among these three stages, the fashion manufacturing stage uses the lowest smart factory solution [29].

The fashion brand business utilizes big data and AI to enhance forecasting accuracy in strategy establishment and decision-making and strengthen the capabilities of market and consumer analysis. These are smart factory solutions referred to as "product traceability systems" and "algorithm of forecasting" (SMEs ma-turity level 4) [30]. However, these solutions are available only to global companies. The most used examples by general fashion manufacturers are 3D computer-aided de-sign (CAD), CLO, virtual and augmented reality mirror, and design custom system, which are difficult to introduce into small fashion brand businesses (SMEs maturity level 3) [31].

**Table 1.** Technology and level of maturity in current fashion smart factories.

| Value Chain | Maturity | Effect | Example |
|---|---|---|---|
| Development (fashion brand business): plan, design/development | Level 3 or 4 | Satisfy customer needs <br> ➢ Improve demand forecasting accuracy <br> ➢ Strengthening consumer and market analysis capabilities | 3D CAD, CLO, VR/AR mirror, 3D virtual fitting, Design custom system, Smart pattern coordination service Product traceability system Algorithm of forecasting |
| Manufacturing (fashion manufacturing business): sourcing/preproduction/ production | Level 1 or 5 | Improve productivity <br> ➢ Production schedule management <br> ➢ Improve quality and accuracy <br> ➢ Improve space utilization <br> ➢ Decrease Labor cost <br><br> Reduce lead time | Automatic sewing machine (pocket, numbering, folding, iron, barcode attachment, inspector, button, hanging, grinding, etc.) Smart production system (IoT and modular sewing system, SewBot, gripper robot arm) Made-to-measure service, On-demand apparel manufacturing system |
| Supply chain (fashion retail business): delivery/distribution/CS | Level 5 | Increase distribution efficiency <br> ➢ Delivery speed improvement/Logistics cost reduction <br> Satisfy customer needs <br> ➢ Store management and quick response | Logistics data in machine learning Remote monitoring service |

The manufacturing stage requires manual labor and is currently the most unautomated process. There is no new labor inflow, which causes a shortage of skilled workers and aging. Therefore, the effects of smart factories desired by the fashion manufacturing business are labor reduction, productivity improvement, and time re-duction. In the manufacturing stage, smart product planning, modular design, and remote-control system solutions such as modularization, digitalization, vision-based sewing, and IoT robotics are being developed (SMEs maturity level 5). However, as mentioned, manufacturing is the least automated stage, the smart factory has hardly been launched, and most of the factory automation has not even been built (SMEs maturity level 1) [32]. The device currently being used at this stage has the most basic automatic sewing machine (pocket sewing, iron, barcode attachment, button sewing, hanging, grinding etc.).

Like the fashion brand business, the focus of the fashion retail business in the last stage of the supply chain is customer satisfaction. The fashion retail business is an intensively developed real-time inventory management and logistics tracking are possible by utilizing the data advancement of machine learning technology (SMEs maturity level 5). Therefore, it is necessary to compare and examine the fashion smart factory technologies through other industry analyses based on reviewing the smart factory solution used in each stage and the maturity level.

## 3. Methods

In recent years, network science has become an increasingly popular approach for studying psychological and cognitive structures. It expresses complex systems as networks based on the mathematical graph theory [33,34]. Semantic network analysis, often called "keyword network analysis", interprets phenomena with networks built by linking words and words appearing in the text to create a map of relationships. Rice and Danowski [35] described the essence of a semantic network as an analysis of text to measure the rela-tionships among words. In this study, the semantic network analysis method was used to interpret the linkage patterns between keywords and express the smart factory-related news discourse in a systematic form. The news articles used for analysis were collected

through Google News. Founded in 2002, Google News is the world's largest news aggregator, covering more than 80,000 news publishers worldwide. Google News has a profound effect on the news media industry and people's daily lives. According to Schindler [36], Google News contributes more than 10 billion clicks per month to publishers' websites and by 2018, it had approximately 150 million unique monthly visitors in the U.S. [37]. Using *BeautifulSoup*, data were collected from a Python-based Google News crawler from January 2016 to December 2020, when Adidas' Speedfactory was established, and news began to spread. We searched for news articles using the search term "smart factory." In total, 15,523 news articles were collected over five years from 1737 journals worldwide, including *CNET* (USA), *The Business of Fashion* (UK), *The Guardian* (UK), *WWD* (USA), *Wall Street Journal* (USA), *Forbes* (USA), *The Australian* (Australia), *Buro 24/7 Singapore* (Singapore), *CTIMES* (Taiwan), *Fibre2Fashion* (India), *Times of India* (India), *BellaNaija* (South Africa), *The Bangkok Post* (Thailand), *China Daily* (China), *The Korea Times* (Korea), etc.

Prior to text mining, the text data underwent basic text preprocessing, including tokenization, stop word removal, and lemmatization. In this process, unimportant words, symbols, or characters were removed. Next, frequency, which is one of the classical methods for retrieving information from large amounts of text, was measured. In addition, opinion mining analysis was conducted to provide implications for the improvement, development direction, and future of smart factories. Opinion mining is a computer technique used to extract, classify, understand, and evaluate the opinions expressed in various online news sources, social media commentaries, and other user-generated content [38]. This study included both positive and negative articles on the smart factory by analyzing articles from all the newspapers provided on Google News, not limited to publishers. We conducted sentiment analysis using a sentiment dictionary by Big Liu to present suggestions on future smart factory construction in a way that adapts and develops the positive and resolves the negative. The Bing Liu Lexicon contains approximately 6800 English words categorized as either negative or positive. It also contains mistyped and slang words, to extend the applicability of the lexicon.

To extract the main topics of discussion regarding smart factories in media, a newspaper corpus was generated and CONCOR analysis was conducted to analyze the structure of the relationships between hidden core subgroups related to a particular subject in a complex network cluster. CONCOR analysis is a method of iteratively performing Pearson's correlation analysis to identify the appropriate levels of similarity in groups. The results can discriminate between more meaningful subgroups ("blocks") in networks where the relationships among keywords are relatively clear. Correlations make it easier to diagnose similarities. CONCOR analysis identifies individual blocks that correspond to a set of highly correlated keywords while simultaneously identifying the relationships between blocks [39]. The process of this study is shown in Figure 1.

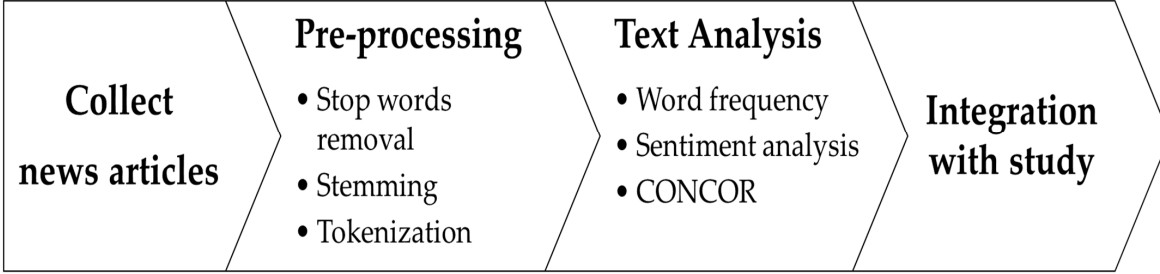

**Figure 1.** Text mining process.

## 4. Results

### 4.1. Semantic Analysis Finding 1: Issue Networks

Table 2 lists the 50 most frequently encountered nouns in the dataset. It provides a general overview of the dominant terms in the discourse of smart factories in news articles.

First, "manufacturing" and "production," the fields that are focused on building smart factories, appeared as key words. Specifically, through the words "car" and "vehicle," it was found that a lot of smart factories were built in the field of automobile manufacturing and production. Words such as "worker," "job," and "people" have also emerged, along with "automation," "robot," "intelligent," and "operation," indicating the direction of the smart factory of unmanned production facilities and automation of management. This shows that the job problems derived from smart factories are also being addressed in the news. On the other hand, there have been many words for Industry 4.0-enabling technologies, such as "datum," "device," "robot," "internet," "sensor," "connected," "IoT," "network," "digital," and "big."

**Table 2.** Top 50 keywords extracted from smart factory-related news.

| Rank | Word | Frequency | Rank | Word | Frequency |
|------|------|-----------|------|------|-----------|
| 1 | Factory | 11,319 | 26 | Business | 562 |
| 2 | Smart | 10,473 | 27 | Process | 540 |
| 3 | Manufacturing | 3866 | 28 | Worker | 525 |
| 4 | Industry | 1941 | 29 | Software | 513 |
| 5 | Plant | 1886 | 30 | Sensor | 510 |
| 6 | New | 1748 | 31 | Vehicle | 497 |
| 7 | Technology | 1638 | 32 | Part | 487 |
| 8 | Company | 1412 | 33 | World | 480 |
| 9 | Automation | 1220 | 34 | Global | 459 |
| 10 | Production | 1206 | 35 | City | 449 |
| 11 | System | 1154 | 36 | Floor | 447 |
| 12 | Car | 894 | 37 | Facility | 446 |
| 13 | Device | 890 | 38 | Connected | 247 |
| 14 | Future | 878 | 39 | IoT | 415 |
| 15 | Datum | 877 | 40 | Network | 413 |
| 16 | Machine | 846 | 41 | Power | 408 |
| 17 | Solution | 807 | 42 | Vision | 408 |
| 18 | First | 721 | 43 | Job | 405 |
| 19 | Robot | 703 | 44 | Way | 404 |
| 20 | Manufacturer | 668 | 45 | Digital | 400 |
| 21 | Internet | 666 | 46 | Big | 390 |
| 22 | Product | 625 | 47 | Intelligent | 390 |
| 23 | Robotic | 611 | 48 | Service | 382 |
| 24 | Thing | 606 | 49 | Application | 375 |
| 25 | Market | 565 | 50 | Operation | 372 |

Meanwhile, the words "world," "global," and "city" indicate that the transition to smart factories is a global trend, and that smart factories are being built as one of the policies for implementing smart cities. Suvarna et al. [40] revealed that smart factories can become key drivers of smart cities by actualizing some key functions such as digital technology, smart citizenship, government initiatives, environment, health and safety, and urban planning.

*4.2. Semantic Analysis Finding 2: CONCOR Analysis of News Articles Related to Fashion Smart Factory*

The results of the CONCOR analysis provided four themes based on frequent words (Figure 2). The four themes of smart factories are listed based on the frequency of the key words: "smart factory construction status," "smart factory development direction," "technologies applied to smart factory," and "features of smart factory." The most frequently mentioned topic in news articles on smart factories was smart factory construction status. Words like "future," "digital," "connect," "world," "global," "intelligent," "manufacturing," "operation," and "management" show that the construction of smart factories is progressing all over the world and that a smart factory that automates, is unmanned, and digitizes manufacturing, operation, and management is being built. On the other hand, words

such as "robotic," "automation," and "datum" show the characteristics of smart factories. Production automation is possible through robots, in which automation must be built on the data. Nowadays, in building smart factories, advanced technologies, such as pattern recognition, deep learning, and unstructured analysis, are used to even analyze the back side of the data. Therefore, the system can not only automatically diagnose problems but also predict future problems and optimization is possible. Technologies that enable such smart factories include the Internet, robot, sensor, and software. It is very important for smart factories to monitor and manage data in real time, such that data can be communicated across the manufacturing chain from manufacturing facilities to multiple material and component partners via the Internet. A variety of sensors are also being used in smart factories, for various purposes ranging from measuring temperature and pressure in industrial processes to monitoring vibrations with equipment and tracking the location of critical assets. In the future, smart factories will not be developed by individual countries or industries, but as a part of a smart city or smart country construction project, as described above, across all industries. Therefore, fashion smart factories should also be built in the direction of automating the manufacturing process by digitizing production plans, work orders, production results, quality information, and introducing automated machines for each product and process. In addition, it is necessary to upgrade management by collecting data and analyzing big data to identify the cause of product failure, machine failure, production delay, and identify the location of workers and products through smart sensors. Software such as the manufacturing execution system (MES), point of production system, and ERP systems can help automate and advance management.

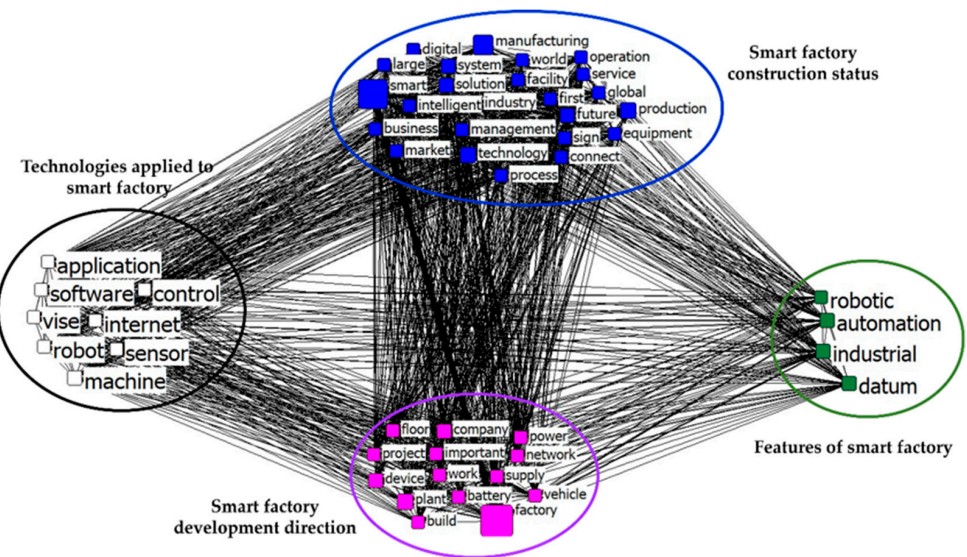

**Figure 2.** CONCOR analysis of second generation of smart factory.

### 4.3. Semantic Analysis Finding 3: Opinion Mining

To provide insights that can help improve and develop smart factories, the emotions expressed in the news articles collection were analyzed and the results expressed in the word clouds are the same as in Figure 3. Word clouds are widely used to summarize and visualize text content and themes. They utilize a variety of font sizes and colors to display important words in a text. They are effective at visually summarizing the main contents of the text and intuitively guiding users to navigate the theme of the text.

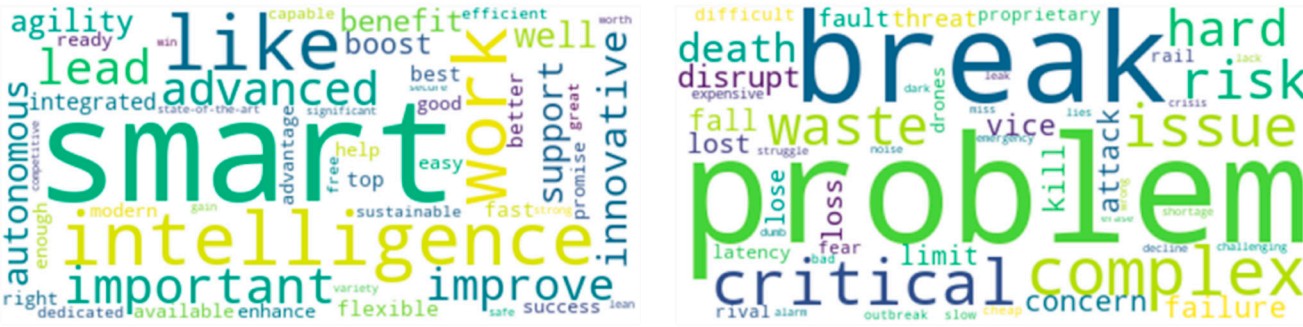

a. Positive emotion related to smart factory　　b. Negative emotion related to smart factory

**Figure 3.** Word clouds of the emotions extracted from the news articles on smart factory.

Positive-emotion words related to smart factory included "smart," "intelligence," "advanced," "innovative," "support," "agility," "fast," "flexible," "easy," "modern," "enhance," "sustainable," "efficient," "state-of-the-art," "competitive," "safe," and "variety." These words show that there is a positive perception in the media that smart factories are not only efficient, fast and safe, but also are collection of state-of-the-art technologies that can perform a variety of functions. Through words such as "sustainable," it was found that smart factories have a positive effect on the creation of sustainable values by companies while pursuing environmental, social, and economic goals. As Kiel et al. [41] observed, smart factories contribute to the current trend of ecological sustainability in production. Furthermore, they can reduce greenhouse gas emissions through data-driven carbon-emission analyses [42]. In addition, the impact of smart factories on reshoring is to relocate production closer to where products and production equipment are purchased, reducing transportation distances and providing environmental, social, and economic benefits [43].

On the other hand, there were negatively valanced words stating that there was a limit to the role a smart factory can play, and that it can cause material damage or time loss owing to breakdown or malfunction. The words "hack" and "virus" indicated the necessity of anticipating the threat of hacking when building a smart factory. In addition, the word "expensive" indicates that building a smart factory is so prohibitive that small and medium-sized enterprises cannot readily afford to; this is a problem we must solve in the future. Fashion smart factories, unlike other industries, have many small-sized companies, so their problems should be carefully looked into. According to Henning [44], although Industry 4.0, as part of a smart, networked world, can increase efficiency and flexibility as well as decrease time-to-market and excess production, it is also closely related to changing workforce qualifications, data security concerns, and expiring business models.

## 5. Conclusion and Discussion

Research on smart factories, along with research on technological applications, such as IoT, CPS, and smart grids, has been conducted using various approaches in various industries such as food [45], cars [46], and shipbuilding [47]. However, there are not many smart factory studies on the fashion industry. Furthermore, the media plays an important role in shaping public opinion and policy making. There is a need to study of smart fashion factories based on related discourse in the media. Moreover, analysis with big data has rarely been conducted in smart factory studies. The purpose of this study was to analyze the development degree and direction of global smart factories through automatic content and semantic network analysis based on more than 15,000 news articles. In addition to making suggestions on the future development directions and policies for the development of the fashion manufacturing sector.

The findings of this study were as follows. First, it can be inferred from words that appear frequently in related news articles that smart factories are being actively adopted in the manufacturing and production fields. In addition, along with the development

direction of smart factories such as automation, unmanned, and intelligent, words such as jobs, workers, and people appeared frequently, indicating that we need to pay attention to the people whose jobs are threatened by the development of smart factories. In other words, as it was an analysis of news articles, both high-tech words about smart factories and negative words about social problems were analyzed. This is because, like the results of previous studies [12–15], news written in the form of a report on the application of advanced technology and news of social criticism were also used for keyword analysis. Through this, it can be confirmed that positive aspects exist about the advanced technology of smart factories, but there are also many negative viewpoints in public opinion. However, looking at the ranking, we found that words related to *job* were lower than those related to *technology*, and the word *global* also appeared, indicating that public opinion on a paradigm shift toward smart factories can be formed in the future. Therefore, it can be confirmed that although the fashion industry is labor-intensive and of low-maturity, it is indispensable to prepare for smart factories.

Next, the CONCOR analysis observed four themes: construction status of smart factory, smart development direction, technologies applied to smart factories, and features of smart factories. These results are about the current smart factory construction and technology, and specific details can be confirmed. They show what needs to be developed and applied carefully compared to the current technology of the smart fashion factory discussed above. In the fashion industry, very advanced technology has appeared in Adidas and Amazon, but most SMEs are at maturity level 1. Most of them use irons, sewing machines, barcode attachments, and so on, and basic smart factory solutions such as devices, software, and networks are hardly present [11]. Only when these basic solutions are built, can fashion manufacturing step forward and undergo a paradigm shift. In addition, after this foundation is built, if robots, data, sensors, and cloud gradually access higher smart factory solutions, it will develop to the final stage. However, it is still at a low level, so a systematic strategy should be established for smart fashion factories.

Finally, the opinion mining of articles related to smart factories reveal advantages such as efficiency, safety, speed, and ecological sustainability. Building a smart factory through these articles helps to form public opinion that various advantages exist. Mostly, cost reduction and usability approaches have been investigated in previous studies [12–15], but this study focuses on sustainability. There, at a time when public interest in sustainability is high due to the COVID-19 pandemic and abnormal weather conditions, the media will continue to mention the advantage of the sustainability of smart factories and help shape positive public opinion. Such public opinion can have a positive effect on the establishment of related policies in the future. On the other hand, through words such as viruses and hacks, smart factories can be seen as threats to cyber security. The shortcomings recognized by the media or the public were more about security than job problems or costs. Therefore, improving the negative security concerns to build a smart fashion factory could reduce the public's negative view. In other words, if news about security improvement appears, negative public opinion will decrease, and positive views and public opinion will be formed, which will greatly influence government policy-making. This can help SMEs in desperate need of government support to build smart fashion factories.

The implications of this study are as follows. First, it has a practical implications for the relevant industry. We examined the extant fashion smart factory solutions and examples, according to the value chain of the fashion industry. In previous studies, most of the necessary technologies, implements, and driving factors were analyzed and presented. However, in this study, we looked at the maturity levels along the value chain of the fashion industry and the technologies currently in use. Through this, we could understand what technology is needed in the future and how to approach the smart factory solution. Therefore, companies that offer smart factory solutions will be able to provide customized solutions according to the various needs required by the companies. Second, it can help the fashion industry's smart factory policy establishment strategy. In the past, extensive big data analysis of smart factories, mostly revealed the necessary technologies on the

topic. However, this study shows how to form a more favorable public opinion, and reduce negative public opinion. Because the fashion industry consists of SMEs, it can be built with government support. Therefore, this study provides guidelines for the direction of government policy establishment. Third, there are also the following academic implications. Prior research has focused on technology development and application. Some studies have addressed risk and benefit in terms of cost. However, to our knowledge, no studies have examined public opinion in the news. Public opinion plays a vital role in various fields. This is because it has a great influence on government policy formation and, corporate strategy establishment. The power of the public and the power of public opinion will continue to increase in the future. Therefore, this study is also meaningful as it is the first attempt to analyze public opinion in a research field where technology is prioritized. This may be helpful for follow-up studies in other technical research fields. Finally, an approach was attempted using a new methodology not often used in smart factory research. The development status and direction of the global smart factory were investigated using newspaper articles from around the world on Google News. In the past, most of the technical analysis of the smart factory or other relevant research was conducted through interviews with related parties. However, it was possible to derive different results through extensive data analysis. In other words, it was necessary to examine the contents of large-scale news articles to understand smart factories better. Therefore, future technology analysis, will help identify groups, policies, and public opinion requiring technology by using extensive data analysis. Furthermore, this study is meaningful in that it drew various viewpoints, such as positive and negative public opinion and advantages and disadvantages of smart factories by analyzing more than 15,000 articles from 1737 journals worldwide with various levels of smart factory development. However, a comparison of smart factory construction conditions and attitudes by country and industry was not conducted. The fashion industry has built an optimal production network by performing each stage of production in various countries worldwide to manufacture products of competitive cost and quality. Therefore, there are differences in countries leading each stage, such as R&D, design, production, marketing, and branding in the manufacture of fashion products, and there are differences in smart factories suitable for each country. Therefore, if a study comparing the elements necessary for building a smart factory and education that can activate smart factory construction is conducted by industry, country, and production stage in the future, it is thought can help build a smart fashion factory.

The limitations of this study are as follows. First, it is difficult to generalize the results of this study to the maturity stage of international smart fashion factories, given that there are few prior research materials outside Korea. Korea is a country that once had a vibrant sewing industry; furthermore, it is making various efforts to pay attention to the sewing industry and cope with the current decline. Therefore, this data exists. However, such research has not been conducted in other countries, resulting in a lack of data. Therefore, if data on the maturity of fashion smart factories are conducted in China or other countries in the future, more reliable results can be achieved. Furthermore, the current status of the construction and development of smart factories around the world was analyzed in this study; however, the study could not be conducted by industry or country. Especially in the fashion industry, only a few global companies are adopting smart factories; therefore, there is not enough data to analyze. Future attempts by the media to analyze smart fashion factory reports will yield more reasonable results.

**Author Contributions:** S.-E.L., N.J. and K.-H.L. have worked closely together on conceptualization, methodology, formal analysis, investigation, data curation, original draft preparation, and writing and editing. All authors have read and agreed to the published version of the manuscript.

**Funding:** This research received no external funding.

**Institutional Review Board Statement:** Not applicable.

**Informed Consent Statement:** Not applicable.

**Data Availability Statement:** Data available on request due to restrictions eg privacy or ethical.

**Conflicts of Interest:** The authors declare no conflict of interest.

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
