# Peer review of "Visioning the Future of Smart Fashion Factories Based on Media Big Data Analysis"

_applsci, doi:10.3390/app11167549_

Round 1

Reviewer 1 Report

Dear authors,

I was pleased to read your article. Following the review of the paper with code applsci-1301577 and entitled “Suggests on the Future of Fashion Smart Factory through Media Big Data Analysis" for Applied Sciences Journal, I mention the following aspects.

In general, the quality (including novelty, value of contribution, language, presentation style, etc) is low for a typical journal paper. Paper (Type: Article) does not really bring to bear any significant new scholarly contributions and the proposed methodology does not appear to offer any improvement on current procedures.

It emerges the need to highlight the associated research question in the introduction section. When deepening your theoretical argumentation, you may in every section, start from a general standpoint before focusing on the specific context.

Authors should focus more on the need of the study, the novelty of this work, and the choice of the methodology. Future research opportunities should also be better emphasized.

It is necessary to emphasize the boundaries of research and the conditions of use of the selected method. For example: what period the analyzed articles come from; who is the author of the analyzed articles; from which country are the analyzed contributions; etc. The correct delimitation of the research problem, the definition of the state of Is and should Be, also determines the quality and usability of the output.

Extend the chapter "Results" about a precise description of the experimental results, their interpretation, as well as the experimental conclusions that can be drawn.  The conclusion could be extended about the possibility of a next specific application of the proposed solution in practice. Future research directions may also be highlighted. In the manuscript, please emphasize the novelty of your approach.

Overall, however, the article is interesting. I look forward to your edited article soon.

My review is not a critique, it's just my recommendations and my view of the post. Finally, I appreciate your effort.

Kind regards

Reviewer

Author Response

Thank you for your valuable comments. Please see the attached response note.

Reviewer 2 Report

The paper is devoted to the problem of analyzing newspapers to suggest the future direction of development of an underdeveloped smart factory of the fashion industry. The study analyzed 15,523 news articles using CONCOR semantic network analysis and analysis methods. The research is aimed at forming an understanding of the current state, direction of development, and vision of the future of the "smart fashion factory".

Main comments on the structure of the paper:
1. In the Introduction, it is necessary to concretize the concepts - motivation, research goal, research questions (which the authors named the research purpose), method, and main contribution of research.
2.The Theoretical Background should end with an indication of the knowledge gap that the study should fill.
3. It is desirable to end the Method section with a diagram showing the sequence of steps performed in the study according to the proposed methodology.
4. Section Conclusion & Discussion should be expanded by discussion of the results obtained with the findings of existing studies. And also it is necessary to formulate more clearly: the methodological, theoretical, and practical contributions of this research.

Author Response

(The authors gave the same response as above.)

Round 2

Reviewer 1 Report

Dear authors,

Thank you for your comment and edit article. The manuscript has significantly improved as compared to the previous version.

Indeed the authors made an effort to improve it and the main weaknesses are solved.

I wish you good luck in further research.

Kind regards,
Reviewer

Author Response

Dear reviewer,

Thank you for giving us the opportunity to submit a revised manuscript. We appreciate the time and effort that you have dedicated to providing your valuable feedback on our manuscript. 

Sincerely,